# Clinical Outcomes with Targeted Temperature Management (TTM) in Comatose Out-of-Hospital Cardiac Arrest Patients—A Retrospective Cohort Study

**DOI:** 10.3390/jcm11071786

**Published:** 2022-03-24

**Authors:** Niels T. B. Scholte, Christiaan van Wees, Wim J. R. Rietdijk, Marisa van der Graaf, Lucia S. D. Jewbali, Mathieu van der Jagt, Remco C. M. van den Berg, Mattie J. Lenzen, Corstiaan A. den Uil

**Affiliations:** 1Department of Cardiology, Erasmus MC—University Medical Center, 3015 GD Rotterdam, The Netherlands; c.vanwees@erasmusmc.nl (C.v.W.); marisavdgraaf@outlook.com (M.v.d.G.); l.jewbali@erasmusmc.nl (L.S.D.J.); m.lenzen@erasmusmc.nl (M.J.L.); 2Department of Intensive Care, Erasmus MC—University Medical Center, 3015 GD Rotterdam, The Netherlands; m.vanderjagt@erasmusmc.nl; 3Department of Hospital Pharmacy, Erasmus MC—University Medical Center, 3015 GD Rotterdam, The Netherlands; w.rietdijk@erasmusmc.nl; 4Department of Intensive Care, Haaglanden Medical Center, 2512 VA The Hague, The Netherlands; remco.vd.berg@hotmail.com; 5Department of Intensive Care, Maasstad Hospital, 3079 DZ Rotterdam, The Netherlands; uilc@maasstadziekenhuis.nl

**Keywords:** cardiac arrest, targeted temperature management, intensive care units, mortality, out-of-hospital

## Abstract

Purpose: we evaluated the effects of the shift of a targeted temperature management (TTM) strategy from 33 °C to 36 °C in comatose out-of-hospital cardiac arrest (OHCA) patients admitted to the Intensive Care Unit (ICU). Methods: we performed a retrospective study of all comatose (GCS < 8) OHCA patients treated with TTM from 2010 to 2018 (*n* = 798) from a single-center academic hospital. We analyzed 90-day mortality, and neurological outcome (CPC score) at ICU discharge and ICU length of stay, as primary and secondary outcomes, respectively. Results: we included 798 OHCA patients (583 in the TTM33 group and 215 in the TTM36 group). We found no association between the TTM strategy (TTM33 and TTM36) and 90-day mortality (hazard ratio (HR)] 0.877, 95% CI 0.677–1.135, with TTM36 as reference). Also, no association was found between TTM strategy and favorable neurological outcome at ICU discharge (odds ratio (OR) 1.330, 95% CI 0.941–1.879). Patients in the TTM33 group had on average a longer ICU LOS (beta 1.180, 95% CI 0.222–2.138). Conclusion: no differences in clinical outcomes—both 90-day mortality and favorable neurological outcome at ICU discharge—were found between targeted temperature at 33 °C and 36 °C. These results may help to corroborate previous trial findings and assist in implementation of TTM.

## 1. Background

The risk of death in comatose patients after out-of-hospital cardiac arrest (OHCA) remains high. Since 2005, one of the cornerstones in the treatment of these patients has been targeted temperature management (TTM) [1]. However, over the years there have been a lot of debate about at what temperature TTM should be performed. The first randomized controlled trials to show a positive effect on mortality and better neurological outcome were carried out using TTM at 33 °C (TTM33) [2,3]. Subsequently, TTM between 32–34 °C were implemented in clinical guidelines [1]. In 2013, a randomized controlled trial carried out by Nielsen et al. [4] (TTM1-trial) showed that the outcomes (mortality and neurological outcomes) with TTM at 36 °C (TTM36) did not differ compared to TTM33. After publication of this study, international guidelines were adapted to TTM between 32–36 °C [5]. However, since the publication of the TTM1 trial, several observational studies have reported potentially higher mortality rates and worse neurological outcomes [6,7,8]. These potentially worse outcomes might have been related to the fact that fever occurred more often and that poorer adherence to TTM was observed after altering the guidelines [6,8]. The recently published TTM2 trial, conducted by the same research group as the TTM1 trial, showed that hypothermia between 32–34 °C had no benefit in outcome over fever prevention alone [9]. Recently, a critical note review was published, in which the high mortality rate in the TTM2 trial was compared to patients treated similarly in other studies and registries as discussed [10].

This shows the significance of presenting other clinical observational outcome data of comatose cardiac arrest patients admitted to the ICU before and after the implementation of TTM36 in our center. After our local guidelines were altered, we started to collect data to evaluate this change over time. For this reason, this study was set up to evaluate the impact of adopting TTM36 versus TT33 on 90-day mortality and neurological outcomes at ICU discharge in our center.

## 2. Materials and Methods

### 2.1. Study Sample, Setting, and Design

We performed a single-center retrospective cohort study of all comatose (GCS on arrival <8) OHCA patients treated with TTM at the Erasmus University Medical Center, Rotterdam, the Netherlands between 1 January 2011 and 31 December 2018. All patients admitted to the ICU had a goal temperature of 33 °C or 36 °C for 24 h by protocol. Prior to 1 July 2016 the patients’ core temperature target was 33 °C (TTM33 group), whereas after this date the target temperature was changed to 36 °C (TTM36 group) according to the changed guidelines. All included patients received TTM with a femoral intravascular cooling device (Thermoguard XP, ZOLL Medical Corporation, Chelmsford, MA, USA). Fever was prevented in the first 72 h in patients who did not regain consciousness. Temperature was measured centrally (every hour) via a temperature probe in the bladder. Patient data was extracted from the electronic patient record and stored in a secured database that is open for reuse. The data extracted consists of demographic characteristics, medical history, arrest characteristics, ICU characteristics, and clinical outcomes. The study was approved by the medical ethics committee of the Erasmus MC (MEC-2019-0206).

### 2.2. Baseline Characteristics and Clinical Outcomes

For baseline characteristics, we collected demographic characteristics (i.e., age and gender), medical history (e.g., cardiac risk factor and history of cardiovascular disease), arrest characteristics (e.g., witnessed arrest and bystander CPR), and post arrest characteristics (e.g., hemodynamic measurements, use of vasoactive agents, and use of mechanical circulatory support) In addition, the lowest temperature during TTM was collected, which was estimated in part by automatic retrieval from the patient data monitoring system (PDMS) by the IT department of Erasmus MC. For clinical outcomes, we collected the cerebral performance category (CPC) score, ICU length of stay (ICU LOS), and 90-day mortality.

### 2.3. Primary and Secondary Outcomes

The primary outcome of this study was 90-day mortality after ICU admission. The secondary outcome was favorable neurological outcome at ICU discharge (yes or no), CPC score at ICU discharge (as a continuous measure) and ICU length of stay (LOS). Favorable neurological outcome consisted of CPC scores 1 (good cerebral function/ minor disability) and 2 (moderate disability), whereas poor neurologic outcome consisted of CPC scores 3 (severe disability), 4 (vegetative status), and 5 (dead). CPC score at ICU discharge can thus range from 1 to 5. The CPC score was based on the ICU discharge report of an attending neurologist and was re-assessed by researchers managing the database.

### 2.4. Statistical Analysis

We analyzed the data in three steps. First, we present the baseline characteristics. Categorical variables are presented as numbers with percentages. Continuous variables are presented as medians with interquartile ranges (IQR). Differences between the two targeted temperature groups were analyzed with a Chi-square test or Fisher’s exact test when the number of patients was lower than five in the cross-table. Differences in continuous variables were analyzed with the Mann–Whitney U test. The Kaplan–Meier curve for 90-day mortality was calculated for both TTM groups and differences were assessed by the log-rank test. A multivariate Cox proportional hazard regression was used comparing the two TTM groups. The model was adjusted for baseline and arrest characteristics that were significantly different (*p* < 0.05). Significant baseline and arrest characteristics with more than 15% missing date were not used in the adjusted models. The result of the Cox proportional hazard regression is expressed as hazard ratio (HR) with 95% confidence interval (95% CI). For the secondary outcome, favorable neurological outcome (favorable CPC is a CPC score of 1 and 2 coded as 1), a binary logistic regression was used, adjusting for significantly different baseline characteristics. These results are presented as odds ratios (OR) with 95% CI. For CPC score at ICU discharge (as a continuous variable) and ICU LOS, a linear regression model was used comparing the two TTM groups. Results are expressed as beta with the associated 95% CI.

We performed a pre-defined subgroup analysis in those patients who had a known lowest reached temperature. The primary analysis also included patients in whom the actual temperatures were not always stored into the PDMS. In this subgroup analysis we investigated whether there was an association between TTM strategy according to the time period and 90-day mortality, favorable neurological outcome, CPC score, and ICU LOS. Statistical analysis was performed using SPSS software, version 25.0. *p* values < 0.05 were considered as statistically significant.

## 3. Results

### 3.1. Study Sample

A total of 1096 OHCA patients were reviewed for inclusion. Of this group, 298 patients were excluded as these patients did not receive TTM (*n* = 267), received extracorporeal cardiopulmonary resuscitation (ECPR) (*n* = 10), or the location of arrest was unknown (*n* = 21). The final sample consisted of 798 patients, of whom 583 were treated in the TTM33 group and 215 patients were treated in the TTM36 group. Figure 1 presents the lowest temperature reached during TTM over the study period. The data has also been used in a previous study on sex differences published by our group [11].

### 3.2. Descriptive Statistics: Baseline Characteristics

Patient characteristics of both groups are presented in Table 1. Patients had a median age of 63.8 (IQR: 53.8–72.2) years and were less often female (23.9%). Witnessed arrest was seen more often in the TTM33 group than in the TTM36 group (78.3% vs. 70.9%, *p* = 0.03, Table 1, arrest characteristics). Post-arrest care characteristics, shown in Table 2, show that bradycardia during TTM occurred more often in the TTM33 group than in the TTM36 group (60.2% vs. 39.2%, *p* < 0.01). Lowest mean arterial blood pressure was lower in the TTM33 group (57 mmHg, IQR54–63 mmHg) than in the TTM36 group (59 mmHg, IQR 54–63 mmHg). A comparison between 90-day survivors and non-survivors was made and is shown in Appendix A.

### 3.3. Primary Outcome: 90-Day Mortality

Our univariate analysis, in Table 2, shows that there is no significant difference in 90-day mortality between the TTM33 and the TTM36 group (39.3% vs. 43.3%, *p* = 0.31). Figure 2 shows a Kaplan–Meier curve comparing 90-day mortality between TTM33 and TTM36 groups. The log-rank test also shows that time to 90-day mortality is not significantly different between the two TTM groups (*p* = 0.37).

We included hypercholesterolemia, previous ICD implantation, and witnessed arrest in our multivariable cox regression model. Appendix A shows that, in the multivariable adjusted cox regression, there is no significant difference between the TTM33 and the TTM36 groups regarding time to 90-day mortality (HR 0.877, 95% CI 0.677–1.135, with TTM36 as the reference group).

### 3.4. Secondary Outcome: CPC Score and ICU Length of Stay

Univariate analysis showed no significant difference in favorable neurological outcome at ICU discharge between both TTM groups (*p* = 0.13; Table 2). Appendix A shows a binary logistics regression model for favorable neurological outcome. The binary logistic regression reports no significant difference between the TTM33 and TTM36 group and favorable neurological outcome at ICU discharge (OR 1.330, 95% CI 0.941–1.879). Also, no significant difference was found between both TTM groups and CPC score (as a continuous measure) at ICU discharge in the linear regression model (Beta −0.128, 95% CI −0.421–0.166). Besides that, a significantly longer ICU LOS was found in the TTM33 group as compared to the TTM36 group (Beta 1.180, 95% CI 0.222–2.138).

### 3.5. Subgroup Analysis

In this study a subgroup analysis was performed, in which only data on patients in whom lowest temperature was available were used. In Appendix A, we show that there was no difference between the TTM33 and TTM36 groups with respect to 90-day mortality (HR 0.853, 95% CI 0.652–1.117) in the Cox regression analysis.

There was no significant difference between TTM groups in explaining favorable neurological outcome at ICU discharge (OR 1.342, 95% CI 0.926–1.946) and CPC score at ICU discharge (Beta −0.124, 95% CI −0.443–0.194). We found a significant longer ICU LOS in the TTM33 group (Beta 1.104, 95% CI 0.121–2.087).

## 4. Discussion

We evaluated how the implementation of the targeted temperature management at 36 °C affects mortality, ICU length of stay, and neurological outcomes in OHCA patients in our hospital. The main goal of TTM is to have a neuroprotective effect by decreasing the cerebral metabolic rate for oxygen. This results in the decrease of the release of harmful amino acids and the production of free radicals after cardiac arrest [12]. However, no difference has been found in the inflammatory effect between TTM33 and TTM36 [13]. What is known is that the TTM33 strategy patients are more often hemodynamically unstable [14,15]. Therefore, it is important to understand whether a difference in clinical outcomes can be found between both TTM strategies.

After changing the local hospital protocol to use TTM at a target temperature of 36 °C instead of 33 °C, our study finds no difference in 90-day mortality between these two strategies. This result found in our study is similar to the results of the study by Nielsen et al. [4] and several recent studies examining differences between the two TTM groups [16,17]. However, several other observational studies have shown a trend towards higher mortality in the 36 °C group [6,7,8]. This can be due to the poor adherence to TTM in these studies. Unfortunately, we are not able to demonstrate in more detail if patients in the TTM36 group received adequate TTM in our study.

For the secondary outcome we found no significant difference between the TTM33 group and the TTM36 group regarding favorable neurological outcome and CPC score at ICU discharge. This result is in line with the result of the study carried out by Nielsen et al. [4]. In recent years, two other studies have been carried out, one of which shows the same results [18]. In contrary, the HYPERION study showed a higher percentage of favorable neurologic outcomes among patients with a non-shockable cardiac arrest treated with mild hypothermia (33 °C) [19]. In this trial, the temperature of patients in the normothermia group was maintained between 36.5–37.5 °C. This strategy is almost identical to what was done in the TTM2 trial; however, this trial found no difference in neurological outcome [9].

We found a longer ICU LOS in the TTM33 group, which is in line with the results found in the study of Salter et al. [7]. We speculate that this difference is due to a longer time of rewarming the patient since the absolute difference in days is low. In addition, the overtime changes in logistics, such as the moment of stopping sedation only during the day and an increased ICU capacity strain resulting in early transfers to other hospitals may also play a role. Another potential explanation for this result might be that patients within the TTM33 group are more deeply sedated and thus it takes longer for those patients to regain consciousness. Also, the delayed clearance of sedatives and neuromuscular blocking agents due to hypothermia may be attributed to the longer ICU LOS. However, in our dataset we have no information regarding sedative medication.

When looking into our ICU characteristics, TTM33 patients more often had bradycardia. It is known that a lower temperature is associated with the occurrence of bradycardia [20]. According to a recent meta-analysis, the occurrence of bradycardia during TTM is a good prognostic factor in OHCA patients [21]. This effect is also seen in our population, as bradycardia occurred more frequently in patients who survived the first 90-days. In the current study, we also found a significant lower mean arterial blood pressure (MAP) in the TTM33 group compared to the TTM36 group. Hemodynamic instability frequently occurs after cardiac arrest and can lead to death due to multi organ failure. Although there is insufficient evidence for a specific hemodynamic goal, the guidelines advise to avoid hypotension (MAP < 65 mmHg) and to target MAP to achieve adequate urine production [12]. Bro-Jeppesen et al. [22] showed a lower cardiac output, which in part determines MAP in patients treated with TTM33. That study showed no statistically significant difference in MAP in TTM33 versus TTM36. However, MAP in that study showed a downward trend over time particularly in the TTM33 group. In another study of Bro-Jeppesen et al. [15], a slightly lower MAP is presented in the rewarming phase of the TTM33 group. It is important to state that in the current study, we found a statistical difference. However, it is questionable if this difference of 2 mmHg is indeed clinically relevant. Finally, we did not have data on the doses of vasoactive medication.

### 4.1. Limitations

Our study has several limitations. First, the retrospective aspect of this study is a limitation, which increases the chance of missing data and the misclassification of different variables compared to a prospective study. However, this is the best available proxy as properly documenting such information at the crucial moments in the arrest is difficult.

Second, CPC scores at ICU discharge were retrospectively determined based on information gathered from patient charts. Although this may introduce some bias, similar methodology has been used by Chocron et al. [23]. Also, at ICU discharge patients can still neurologically improve or deteriorate; therefore, the CPC score at the time of ICU discharge can be an over- or underestimation of the long-term neurological outcome. Unfortunately, this study does not provide information regarding long-term neurological outcome.

Third, we were not able to completely reconstruct the complete temperature paths during and after TTM. Therefore, we have no data regarding the time until target temperature was reached and the occurrence of fever after rewarming.

Finally, in our research we did not focus on the adverse events of targeted temperature. Since the efficacy of both TTM strategies lie closely together, this could implicate that adverse events may be an important factor to take into account when formatting a new guideline regarding targeted temperature. The INTCAR2-study and TTM2 trial were able to present an overview of adverse events in which only hemodynamic instability occurred more often in the hypothermia group [9,24].

### 4.2. Future Research

This study shows that there is no difference in 90-day mortality and neurological outcomes between both TTM strategies. Observational studies such as these are important since they capture real life data outside trial settings and illustrate how easy or difficult implementation of trial results may be. Future observational studies should be done to confirm trial results in routine clinical practice, especially given that the latest data on TTM2-trial in which the investigators reported no difference in 6-months mortality and functional outcome between hypothermia versus normothermia with fever prevention alone [9].

## 5. Conclusions

No differences in clinical outcomes—both 90-day mortality and favorable neurological outcome at ICU discharge—were found between targeted temperature at 33 °C and 36 °C in a population receiving complete intravascular temperature management. These results may help to corroborate previous trial findings and assist in implementation of TTM.

## Figures and Tables

**Figure 1 jcm-11-01786-f001:**
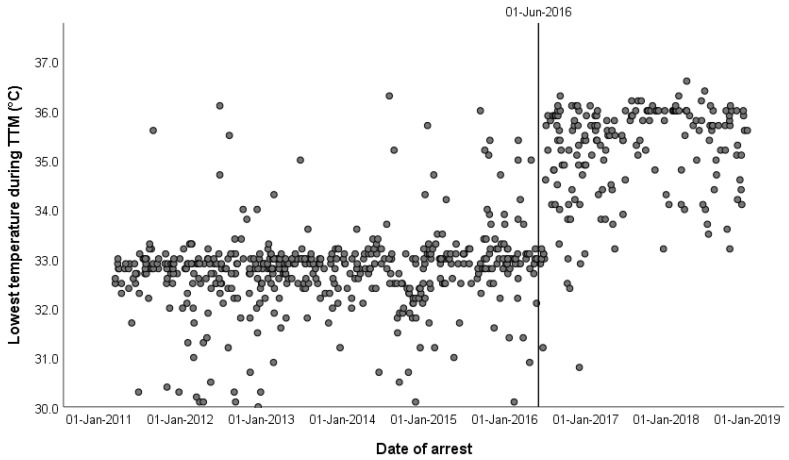
Scatterplot of the lowest temperatures during TTM (vertical line represents the adaption of the TTM protocol). 24/497 (4.8%) of the patients in the TTM33 group had a lowest temperature of 34 °C or higher.

**Figure 2 jcm-11-01786-f002:**
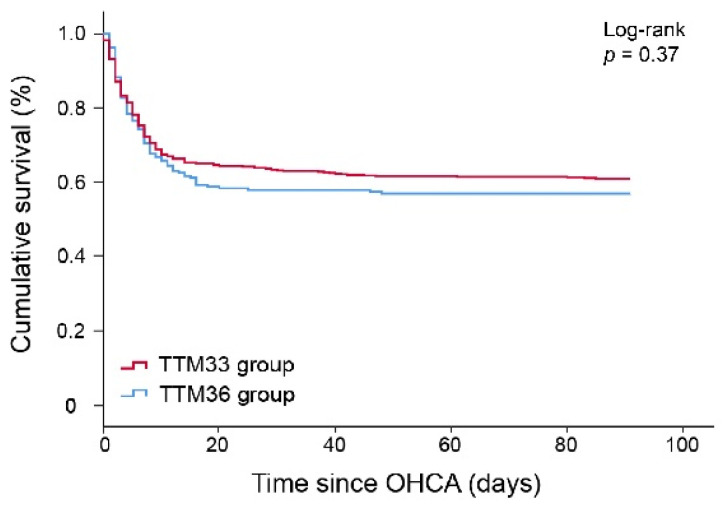
Kaplan–Meier curve 90-day mortality.

**Table 1 jcm-11-01786-t001:** Patient and arrest characteristics in OHCA patients admitted to the ICU.

Characteristics	All TTM Patients (*n* = 798)	TTM33 Group (*n* = 583)	TTM36 Group (*n* = 215)	*p* Value	Missing *(n)*
**Demographic**					
Age, years (IQR)	63.8 (53.8–72.2)	63.4 (53.0–71.5)	64.9 (52.9–74.4)	0.30	0
Female, *n* (%)	191 (23.9)	137 (23.5)	54 (25.1)	0.66	0
**Medical history, *n* (%)**					
Hypertension	283 (37.9)	203 (37.9)	80 (38.1)	0.96	52
Hypercholesterolemia	198 (26.6)	157 (29.3)	41 (19.5)	0.006	53
Diabetes Mellitus	151 (19.8)	104 (18.8)	47 (22.4)	0.27	36
Family history of CVD	115 (18.8)	94 (21.4)	21 (12.2)	0.01	186
Smoking	215 (32.7)	167 (35.0)	48 (26.5)	0.04	140
Peripheral vascular disease	66 (8.7)	50 (9.1)	16 (7.7)	0.54	38
Previous myocardial infarction	175 (22.0)	130 (22.3)	45 (21.0)	0.69	2
Chronic heart failure	89 (11.2)	63 (10.8)	26 (12.1)	0.60	3
Previous PCI	116 (14.6)	78 (13.4)	38 (17.8)	13	3
Previous CABG	54 (6.8)	41 (7.1)	13 (6.1)	0.63	3
Previous ICD implantation	21 (2.6)	9 (1.5)	12 (5.6)	0.004	2
Previous pacemaker implantation	19 (2.4)	12 (2.1)	7 (3.3)	0.33	2
Previous TIA or stroke	64 (8.0)	46 (7.9)	18 (8.4)	0.83	1
Pulmonary embolism	11 (1.4)	7 (1.2)	4 (1.9)	0.48	2
**Arrest characteristics, *n* (%)**					
Location of arrest				0.38	0
*Home*	440 (55.1)	316 (54.2)	124 (57.7)		
*Public*	358 (44.9)	267 (45.8)	91 (42.3)		
Witnessed arrest	592 (76.4)	448 (78.3)	144 (70.9)	0.03	23
Bystander CPR	504 (65.2)	363 (64.0)	141 (68.4)	0.25	25
Estimated time to CPR				0.30	127
*0–5 min*	515 (76.8)	380 (75.1)	135 (81.8)		
*6–10 min*	125 (18.6)	100 (19.8)	25 (15.2)		
*11–20 min*	26 (3.9)	21 (4.2)	5 (3.0)		
*>20 min*	5 (0.7)	5 (1.0)	0 (0)		
AED defibrillation	320 (40.3)	232 (39.8)	88 (41.5)	0.66	3
Shockable initial rhythm	596 (78.1)	435 (77.7)	161 (79.3)	0.63	35
Defibrillation by EMS	523 (65.6)	382 (65.5)	141 (65.9)	0.92	1
Time to ROSC				0.07	149
*0–5 min*	33 (5.1)	28 (5.8)	5 (3.0)		
*6–10 min*	116 (17.9)	92 (19.0)	24 (14.5)		
*11–20 min*	297 (45.8)	224 (46.4)	73 (44.0)		
*>20 min*	203 (31.3)	139 (28.8)	64 (38.6)		
Pre-hospital intubation	509 (63.8)	364 (62.4)	145 (67.4)	0.19	0
Primary cardiac cause	693 (92.0)	508 (91.7)	185 (93.0)	0.57	45

Note: Descriptive statistics of the sample comparing the TTM33 group with the TTM36 group. IQR: Interquartile Range, CVD: Cardio-Vascular Disease, CABG: Coronary-Artery Bypass Grafting, PCI: Percutaneous Coronary Intervention TIA: Transient Ischemic Attack, ICD: Implantable Cardioverter Defibrillator, CPR: Cardiopulmonary Resuscitation, AED: Automatic External Defibrillator, EMS: Emergency Medical Service. ROSC: Return of Spontaneous Circulation.

**Table 2 jcm-11-01786-t002:** Post arrest care characteristics and clinical outcomes in OHCA patients admitted to the ICU.

Characteristics	All TTM Patients (*n* = 798)	TTM33 Group (*n* = 583)	TTM36 Group (*n* = 215)	*p*-Value	Missing *(n)*
**Post arrest care characteristics**					
Lowest temperature during TTM, °C (IQR)	33.0 (32.7–34.2)	32.9 (32.5–33.0)	35.6 (34.7–36.0)	<0.001	117
Bradycardia during TTM, *n* (%)	389 (55.5)	327 (60.2)	62 (39.2)	<0.001	97
Lowest MAP during TTM, mmHg (IQR)	57 (52–61)	57 (54–63)	59 (54–63)	<0.001	51
Inotropics/vasoactive drugs, *n* (%)	737 (98.4)	567 (98.1)	170 (99.4)	0.23	49
Mechanical circulatory support, *n* (%)				0.003	0
IABP	94 (11.6)	82 (13.9)	12 (5.5)		
Impella	1 (0.1)	0 (0.0)	1 (0.5)		
ECMO	7 (0.9)	4 (0.7)	3 (1.4)		
CVVH, *n (%)*	37 (5.0)	29 (5.0)	8 (4.7)	0.88	51
**Clinical outcomes**					
Mortality at 90 days, *n* (%)	322 (40.4)	229 (39.3)	93 (43.3)	0.31	0
Neurologic status at ICU discharge, *n* (%)				0.13	26
Favorable neurologic outcome, CPC 1–2	450 (58.3)	342 (59.2)	108 (53.7)		
Poor neurologic outcome, CPC 3–5	322 (41.7)	229 (40.1)	93 (46.3)		
Neurologic status at ICU discharge, *n* (%)				0.002	26
CPC 1—Full neurologic recovery	269 (34.8)	194 (34.0)	75 (37.3)		
CPC 2—Mildly impaired	181 (23.4)	148 (25.9)	33 (16.4)		
CPC 3—Awake with severly impaired neurologic status	29 (3.8)	22 (3.9)	7 (3.5)		
CPC 4—Comatose, unresponsive	23 (3.0)	10 (1.8)	13 (6.5)		
CPC 5—Dead	270 (35.0)	197 (34.5)	73 (34.5)		
ICU length of stay, days (IQR)	4.0 (3.0–7.0)	5.0 (3.0–7.0)	4.0 (2.0–6.0)	0.001	22

Note: Descriptive statistics of the sample comparing the TTM33 group with the TTM36 group. IQR: Interquartile Range, TTM: Targeted Temperature Management, IABP: Inta-Aortic Balloon Pump, ECMO: Extra Corporal Membrane Oxygenation, CVVH: Continues Veno-Venous Hemofiltration, ICU: Intensive Care Unit, CPC: Cerebral Performance Category score.

## Data Availability

The data and analysis scripts are available upon request from the corresponding author.

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
