# Peer review of "Clinical Outcomes with Targeted Temperature Management (TTM) in Comatose Out-of-Hospital Cardiac Arrest Patients—A Retrospective Cohort Study"

_jcm, 2022, doi:10.3390/jcm11071786_

Round 1

Reviewer 1 Report

Dear authors,

You replied to the questions raised by the reviewers. Thank you.

I don't have any further comment

Author Response

Dear reviewer,

We would like to thank you again for your previous comments and suggestions. This helped to improve the manuscript considerably.

Kind regards

On behalf of the (co)authors,

Niels Scholte

Reviewer 2 Report

The authors report that their study found a significant higher mean arterial blood pressure in the TTM36 group. However, the confidence interval for both the groups overlaps. So, unclear how authors are reporting higher mean BP in TTM36 group. Furthermore, they have not added this outcome in their results section, while they have briefly discussed about under discussion (would recommend to add clinical value rather than reporting results of other studies only).

What does 'missing' column mean in the results section? Please clarify.

Authors are applying a clinical trial into their single center setup to compare outcomes for TTM33 and 36. However, authors report that they 'are not able to demonstrate in more detail if patients in the TTM36 group received adequate TTM in our study.' This puts into question the validity of this study.

Also, would recommend to discuss clinical significance of TTM36 vs 33, rather than just stating it in the manuscript.

Author Response

This manuscript is a resubmission of an earlier submission. The following is a list of the peer review reports and author responses from that submission.

Round 1

Reviewer 1 Report

The authors have aimed to evaluate the effect of shift of TTM strategy from 33 to 36°C in OHCA patients admitted to the ICU in a single center.

While TTM1 and TTM2 trial have favored 36°C, the authors aimed to present clinical outcomes in their center before and after implementation of the new guidelines (which state 36°C as the choice for TTM strategy). After multivariate cox regression, authors found no significant difference between the TTM33 and the TTM36 groups regarding 90-day mortality, favorable neurological outcome and longer length of stay in the TTM33 group.

Authors have done a great job in explaining the methods, sub-group analysis, results and the discussion.

Authors report that their study found a significant higher mean arterial blood pressure in the TTM36 group. Would add potential explanations for this in the discussion.

Also, there are some grammatical errors which need to be fixed.

Reviewer 2 Report

Dear authors,

Scholte and coworkers present a retrospective, single-centre, observational study on 2 temperature management strategies after out of hospital cardiac arrest over 2 time periods.

Although the study comprises a quite large cohort with a high proportion of “good” neurological outcome at ICU discharge, it does not add any novelty regarding outcome after OHCA in the light of TTM. Quite a few, cited papers including some randomized clinical trials have already shown this finding. Unfortunately, you are not able to provide the same quality of data, like long term neurological outcome, detailed data on temperature,…

Furthermore, the paper suffers from major methodological flaws, like just relying on the lowest temperature measurement to classify the patients respectively to characterize the patients’ temperature course. You should use mean or median temperatures, as during TTM in whichever group, “outliers to the desired and set temperature” are very common. You do also not mention if this single temperature was just 1 recording or present over a certain time period (hours).

I’m also wondering about the lowest temperature (especially in the TTM33 group) close to 38°C. This must either be a typing error, a major deviation from the protocol (why? because of deemed futility?) or a reflection of incomplete data recording as the paper does not include any information on the timepoint when this “lowest” temperature was recorded. Once again, a mean or median temperature over the first 24 (or even 28 to 36) hours would have been a better choice than merely reporting a single temperature value for classifying patients.

Methodologically, mixing up patients from the initial TTM groups by reclassifying them according to their actual lowest temperature (with all the possible flaws mentioned above) is not correct as this is a selection bias. Probably those in the first period with high “lowest” temperatures are those deemed “not worth” getting TTM33, so in your secondary analysis, these potentially “poor outcome” patients end up with the TTM36 cohort in a whole. Whereas in the second period, those with lower temperatures are put to the TTM33 group, but a low temperature might be a sign of “poor outcome if it is spontaneously low” or the result of the application of TTM33 to a patient that might “benefit” from TTM33. In general, reclassifying patients as TTM33 or TTM36 simply by the time period is obviously also critical as some patients have “lowest temperatures” in the first timeperiod close to 38°C which definitely is not TTM33. Again, the issue with a single (lowest) temperature is a major concern.

Another issue might also be the “highest” measured temperature as fewer is believed to be a determinant for poor outcome. You do not mention this aspect at all.

Minor comments:

The TTM and TTM2 trial both used fixed TTM goals at 33°C (and not between 32-34°C) as you mention.

The sentence line 53ff should be rewritten as it is unclear.

Table 1 write “Defibrillation” instead of Defibliration”

Your discussion about hypercholesterolemia an ICD is, in my opinion, an epiphenomenon, not worth of discussing in this setting. You discuss the mean arterial pressure with vasopressor use, again, as you don’t have any data, you cannot interpret the lower MAP in the TTM33 only based on similar use of vasoactive drugs or inotrope as the doses are not reported. Furthermore, the discussion about the “best or optimal” MAP in OHCA and TTM still is open to debate. So no need to discuss this issue here.

I agree with your limitation that you don’t have data on the timely evolution of temperature, which is the main flaw of this paper, but I don’t agree with your statement that you probably reached target temperature faster than in randomized trials. First, one has always the tendency to overestimate the own performance, second, you don’t provide any data on your performence and third, in your TTM33 period, quite a lot of patients show lowest temperatures well above 34°C.